# Internalization and presentation of myelin antigens by the brain endothelium guides antigen-specific T cell migration

Melissa A Lopes Pinheiro, Alwin Kamermans, Juan J Garcia-Vallejo, Bert van het Hof, Laura Wierts, Tom O'Toole, Daniël Boeve, Marleen Verstege, Susanne MA van der Pol, Yvette van Kooyk, Helga E de Vries*[†], Wendy WJ Unger*[†][‡]

Department of Molecular Cell Biology and Immunology, Vrije Universiteit Medical Center, Amsterdam, The Netherlands

*For correspondence: he.
devries@vumc.nl (HEdV);
w.unger@erasmusmc.nl (WWJU)

[†]These authors contributed
equally to this work

Present address: [‡]Department
of Pediatrics, ErasmusMC-Sophia
Children's Hospital, Rotterdam,
The Netherlands

Competing interests: The
authors declare that no
competing interests exist.

Reviewing editor: Gary L
Westbrook, Vollum Institute,
United States

**Abstract** Trafficking of myelin-reactive CD4[+] T-cells across the brain endothelium, an essential step in the pathogenesis of multiple sclerosis (MS), is suggested to be an antigen-specific process, yet which cells provide this signal is unknown. Here we provide direct evidence that under inflammatory conditions, brain endothelial cells (BECs) stimulate the migration of myelin-reactive CD4[+] T-cells by acting as non-professional antigen presenting cells through the processing and presentation of myelin-derived antigens in MHC-II. Inflamed BECs internalized myelin, which was routed to endo-lysosomal compartment for processing in a time-dependent manner. Moreover, myelin/MHC-II complexes on inflamed BECs stimulated the trans-endothelial migration of myelin-reactive Th1 and Th17 2D2 cells, while control antigen loaded BECs did not stimulate T-cell migration. Furthermore, blocking the interaction between myelin/MHC-II complexes and myelin-reactive T-cells prevented T-cell transmigration. These results demonstrate that endothelial cells derived from the brain are capable of enhancing antigen-specific T cell recruitment.

## Introduction

In the neuro-inflammatory disorder multiple sclerosis (MS), the trafficking of immune cells into the brain is a crucial step not only in the onset but also the progression of the disease (*Lublin et al., 2014*; *Sospedra and Martin, 2005*). Once within the central nervous system (CNS), myelin-reactive T-cells induce severe neuronal and tissue damage and degeneration (*Trapp et al., 1998*). During the initiation of the disease, myelin-specific CD4[+] T cells are differentiated into effector T helper (Th) 1 or Th17 cells (*Bailey et al., 2007*; *Bielekova et al., 2000*). There is compelling evidence that Th1 and Th17 cells, separately or in cooperation, mediate deleterious responses in MS (*Carbajal et al., 2015*).

To enter the CNS, immune cells have to cross the blood-brain barrier (BBB), which is composed of highly specialized brain endothelial cells (BECs) that are sealed by closely regulated tight junctions (*Tietz and Engelhardt, 2015*). Lymphocyte migration into the CNS parenchyma is a multi-step process that requires close contact between lymphocytes and BECs (*Ransohoff et al., 2003*). These cell-cell contacts are mediated by cell surface molecules on both the lymphocytes and BECs. During inflammation BECs upregulate the expression of the adhesion molecules Inter-Cellular Adhesion Molecule (ICAM)−1, Vascular cell adhesion molecule (VCAM)−1 and Activated leukocyte cell adhesion molecule (ALCAM), which are necessary for the firm adhesion of lymphocytes to the endothelium as well as for the trans-migration process (*Larochelle et al., 2011*). Although these adhesion

**eLife digest** The blood vessels in the brain help to control the entry of nutrients, cells and waste products into and out of the brain. In doing so, they create a protective barrier between the blood and the brain known as the blood-brain barrier. However, this barrier loses its protective function in individuals with multiple sclerosis or other disorders that affect the brain. Multiple sclerosis patients develop inflammation and their immune cells become able to enter the brain. These immune cells may then attack layers of insulation called myelin that surround nerve cells. Myelin helps nerve cells to work properly so the loss of this insulation can lead to tissue damage and cognitive problems. When immune cells called T cells enter the brain they can become primed to recognize myelin and attack it in the same way that they would attack viruses or bacteria. However, it is not clear precisely how these T cells develop the ability to cross the blood-brain barrier and attack myelin.

Now, Lopes Pinheiro et al. show that "endothelial" cells in the blood-brain barrier are able to present fragments of myelin to T cells, which enables the T cells to identify myelin and move into the brain. First, the blood-brain barrier cells absorb and break down proteins in the myelin, and then they present fragments of these proteins on their surfaces with the help of protein clusters called major histocompatibility complexes (MHCs). Other protein fragments that can also activate T cells in other parts of the body did not affect the blood-brain barrier when they were presented by MHCs, which suggests that the effect could be specific to myelin proteins.

The experiments also show that it is possible to stop T cells from crossing the blood-brain barrier by preventing them from interacting with myelin fragments presented by MHCs. This suggests that therapies that interfere with the ability of blood-brain barrier cells to break down myelin proteins and present them to T cells might help to protect the brains of patients with multiple sclerosis.

molecules expressed by BECs have been shown important for the transendothelial migration of leukocytes, the complexity of this interaction and the molecules involved remain poorly understood.

Several studies provided evidence for an antigen-specific component in the transmigration process of encephalitogenic T-cells (*Archambault et al., 2005*; *Galea et al., 2007*; *Ludowyk et al., 1992*). Using an adoptive transfer model of murine MS (experimental autoimmune encephalomyelitis, EAE) it was demonstrated that only activated myelin-specific CD4$^+$ T-cells accumulated in the CNS parenchyma, while non-CNS-specific T-cells failed to infiltrate (*Archambault et al., 2005*). Furthermore, expression of MHC-II on the recipient cells appeared to be required for CNS infiltration, as the myelin-specific T-cells did not transmigrate over CNS vascular endothelium when adoptively transferred in MHC-II deficient mice. However, whether the antigen-specific signal was provided by APCs or BECs was not elucidated. Also infiltration of CD8$^+$ T-cells into the brain was shown to be an antigen-specific process: haemagglutinin-specific CD8$^+$ T-cells were only detected in the CNS upon intra-cerebral injection of cognate, but not control, peptides in an haemagglutinin T-cell receptor transgenic mouse (*Galea et al., 2007*). A role for BECs in providing the antigen signal to T-cells was claimed by showing luminal expression of MHC-I on BECs and the fact that intra-venous injection of a blocking MHC-I antibody significantly reduced the CD8$^+$ T-cell infiltration. However, since a soluble, nominal MHC-I epitope was used as antigen, exogenous binding of these peptides to MHC-I molecules on other cells in the CNS, or even in CNS-draining lymph nodes cannot be excluded (*Weller et al., 1996*). Thus, so far no direct evidence is provided for a role of BECs in processing and presenting CNS-derived antigens during inflammatory conditions.

BECs have been shown to express MHC-I while MHC-II is virtually absent. Inflammation induced activation of BECs causes increased expression of MHC-II as well as of the co-stimulatory molecule CD40 and enhanced their ability to stimulate the proliferation of allogeneic T-cells in-vitro (*Wheway et al., 2013*). Although BECs have been shown to take up soluble antigens by macro-pinocytosis and clathrin-coated pits (*Wheway et al., 2013*), not much is known about their capacity to process and present internalized antigens.

We therefore explored the potential of BECs as antigen-presenting cells and determined whether antigen-presentation by BECs contributes to transmigration of myelin-reactive T-cells. We here demonstrate that inflamed BECs take up and process myelin via the endo-lysosomal degradation

pathway in a time-dependent manner. Importantly, these myelin-derived antigens are presented in *de-novo* expressed MHC-II molecules and facilitate the migration of antigen-specific Th1 and Th17 pathogenic T-cells through the brain endothelium. Better insight into the events that trigger T-cell migration into the brain is crucial for our understanding of MS pathogenesis and will aid the development of new treatments to prevent T-cell infiltrating the CNS.

## Results and discussion

### Brain endothelial cells internalize exogenous antigens irrespective of their activation status

To determine if BECs play a role in antigen-specific migration of CD4$^+$ T cells by acting as APCs, we first assessed the expression of molecules necessary for antigen presentation and co-stimulation. Resting, non-inflamed, human BECs express MHC-I and PD-L1 while MHC-II, CD40 and VCAM$-$1 are expressed at low levels (*Figure 1A*). Upon inflammatory activation, BECs express high levels of VCAM$-$1, and significantly increased the expression levels of MHC-II (*Figure 1A,B*). Similarly, CD40 expression was increased upon activation. Both MHC-I and PD-L1 were highly expressed on resting as well as on activated BECs. Expression of the classical co-stimulatory molecules CD80 and CD86 were undetectable on resting and activated BECs (data not shown). Comparable changes in phenotype were observed when BECs were activated using IFN-γ instead of TNFα (*Figure 1—figure supplement 1*) Together, these results confirm and extend previous findings (*Wheway et al., 2013*) and indicate that BECs are equipped to present antigens under inflammatory conditions. Up-regulation of MHC class II molecules via inflammation induced CIITA activity has been associated with increased susceptibility of EAE, yet how increased MHC-II expression contributes to actual disease has so far not been described (*Reith et al., 2005*).

Since myelin-derived antigens are the major target of auto-reactive T-cells in MS, we investigated if BECs can take up and process myelin. We therefore incubated BECs with fluorescent labeled myelin for different time-points under resting and inflammatory conditions and determined myelin uptake by flow cytometry. As depicted in *Figure 1C* a time-dependent increase in the proportion of myelin$^+$ BECs was observed. Moreover, this process is not significantly affected by treatment with inflammatory stimuli as activated BECs showed a similar amount of internalized myelin as resting BECs. Using imaging flow cytometry, we assessed that BECs that were able to capture myelin increased the number of myelin particles over time to a maximum of three myelin particles/cell after a 24 hr incubation (*Figure 1D*). Moreover, the average amount of myelin particles per cell was the same in both resting and inflammatory conditions, again, demonstrating that this process is not significantly affected by treatment with inflammatory stimuli (*Figure 1D,E*). Of note, in order to measure whether the localization of the myelin signal was intracellular or membrane-bound, we designed a mask that excludes the cell membrane and calculated a ratio of the amount of fluorescence located in the mask *vs* the total amount of fluorescence, as previously reported (*Garcia-Vallejo et al., 2015*). The results indicate that the myelin fluorescence signal was intracellular, demonstrating that BECs are able to efficiently internalize myelin (*Figure 1—figure supplement 2*).

### Myelin internalized by BECs is directed to the endo-lysosome compartments

The endo-lysosomes are the typical antigen-processing compartments of APCs (*Blum et al., 2013*; *Roche and Furuta, 2015*). This intracellular route allows optimal processing of exogenous protein antigens and transfer of antigen-derived peptides to the MHC-II compartment for loading and subsequent presentation to CD4$^+$ T-cells. To determine whether internalized myelin is shuttled to these compartments in BECs, myelin-treated BECs were stained with antibodies against EEA1 (a marker of early endosomes) and LAMP1 (a marker of late endosomes and lysosomes) to measure co-staining with myelin using imaging flow cytometry. We observed that myelin co-localized with both EEA1 and LAMP1 as shown by a high co-localization score (*Figure 2A,B*). The co-localization with both markers was higher at 24 hr of exposure to myelin compared to 4 hr. Since the increase of the co-localization score for myelin-EEA1 was not as strong as shown for myelin-LAMP1 at 24 hr (*Figure 2A,B*), this suggests that at that time point the majority of myelin was present in lysosomes. However, non-internalized myelin fragments that are attached to the cell membrane, could

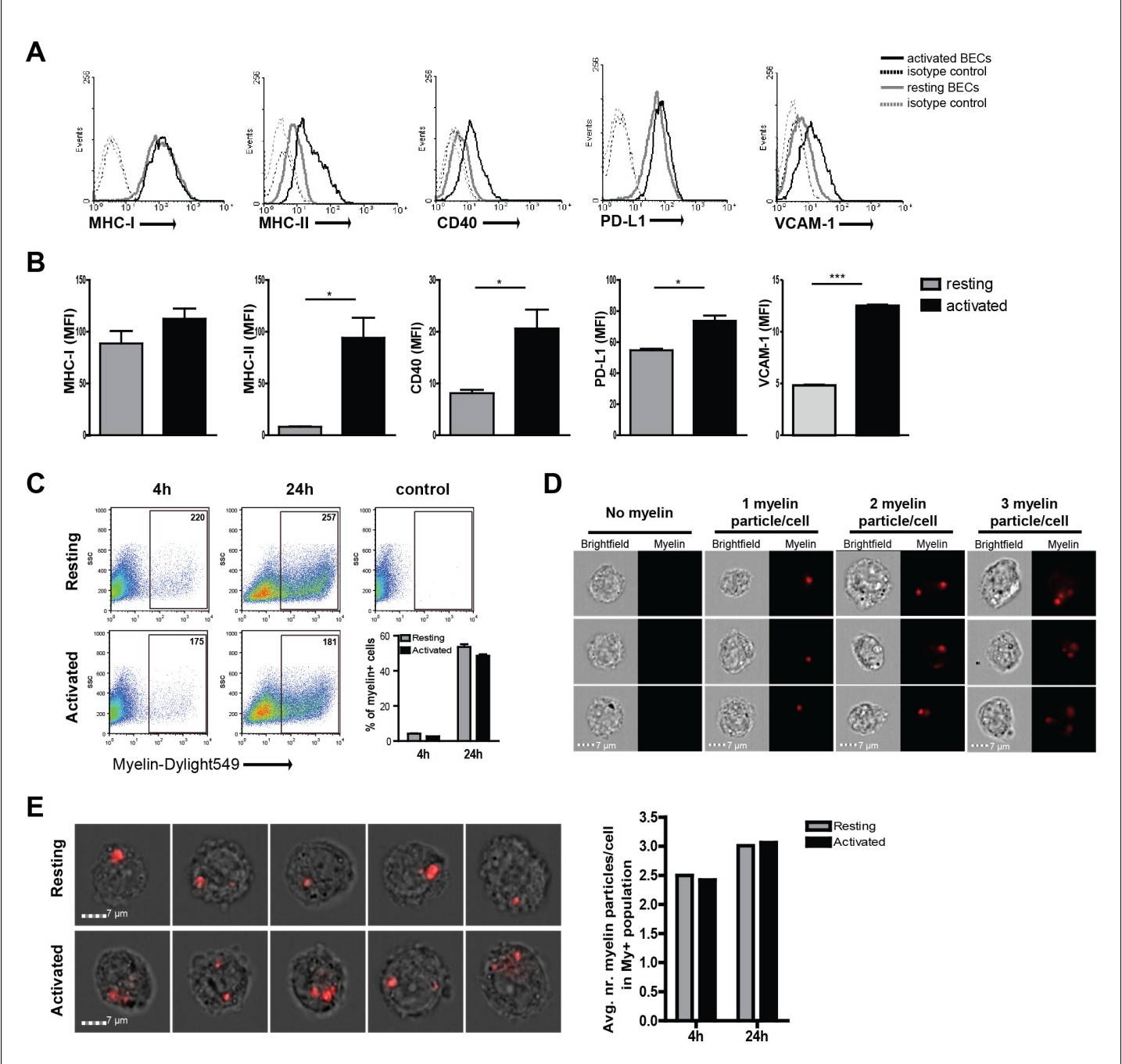

**Figure 1.** Human brain endothelial cells internalize myelin particles. Confluent monolayers of brain endothelial cells (BECs) were stimulated with 5 ng/ml TNFα for 24 hr. (**A**) Expression of MHC-I, MHC-II, CD40, PD-L1 and VCAM−1 was determined by flow cytometry. Histograms depict expression of indicated markers in resting (grey solid line) and activated (black solid line) BECs. Dashed lines indicate isotype controls. (**B**) The MFI of expression of the indicated markers is shown. Data are presented as the mean ± SD of duplicate values (n = 5 independent experiments). *p<0.05, **p<0.01, ***p<0.001 (Student *t*-test). (**C–E**) Fluorescent labeled human myelin was added to resting or activated BECs for 4 hr or 24 hr and uptake was analyzed by (**C**) flow cytometry or (**D–E**) imaging flow cytometry. (**C**) Representative facs plots of myelin uptake by BECs, numbers in plots indicate the MFI of myelin-positive cells. The percentage of myelin-positive resting and activated BECs at 4 and 24 hr after loading with antigen is shown in a graph. (**D–E**) Myelin-positive BECs internalized between 1–3 particles/cell. On average, BECs acquired 2–3 myelin particles/cell. Activation of BECs did not affect the number of internalized particles. The average number of internalized myelin particles per cell is shown in a bar graph. Data presented are the means of triplicate values ± SEM of at least three independent experiments. *p<0.05, **p<0.01, ***p<0.001 (Student *t*-test).

The following figure supplements are available for figure 1:

*Figure 1 continued on next page*

*Figure 1 continued*

**Figure supplement 1.** Human brain endothelial cells increase MHC and costimulatory molecule expression upon activation by IFN-γ.

**Figure supplement 2.** Brain endothelial cells internalize myelin particles.

potentially be 'internalized' as a consequence of trypsinization of adherent BECs. To demonstrate that myelin is actively taken up by BECs, we analyzed myelin uptake and intracellular routing in adherent BECs using confocal laser scanning microscopy. Similar to our experiments using imaging flow cytometry, we observe that 24 hr after loading of adherent BECs a proportion of cells show internalized myelin. Furthermore, it is clear that internalized myelin is present within LAMP1 positive vesicles and not with EEA1 positive organelles (*Figure 2C–I*).

Together, these data suggest that myelin enters the endosomal/lysosomal pathway when internalized by BECs. Furthermore, it is clear that the efficiency of myelin uptake and internal routing to compartments associated with antigen processing by BECs is not affected by inflammation per se. This is in contrast to their professional counterparts: only in an immature state DCs possess high antigen internalization and processing capacities. Activation induced maturation of DCs strongly reduces these functions, and increases the presentation of antigens in MHC molecules (*Inaba et al., 2000*; *Jin et al., 2004*).

## Migration of myelin-specific T-cells depends on presentation of myelin-antigens in MHC-II by BECs

Taking the lack of human myelin-specific T-cell clones as well as HLA-matching issues with BECs into account, we used murine BECs (mBECs) and MOG$_{35-55}$-specific CD4$^+$ T-cells from 2D2 transgenic mice (*Bettelli et al., 2003*) as a model system to elucidate whether myelin antigens are processed and presented by BECs to facilitate T-cell transmigration. Notably, an increased expression of MHC-II was observed on cerebral blood vessels of mice in the active phase of EAE when compared to control adjuvant injected mice (CFA; not shown), demonstrating that mBECs, similar to the human counterparts, are properly equipped to present antigens to pathogenic CD4$^+$ T cells. Since in the brain of MS patients and of EAE mice mainly Th1 and Th17 effector cells have been found (*Carbajal et al., 2015*), we generated MOG-specific Th1 and Th17 in-vitro (*Figure 3A*) and used them in a trans-well setting with myelin-loaded activated mBECs. To allow sufficient antigen processing, mBECs were loaded with myelin in the presence of TNFα 24 hr prior the co-culture with T-cells. To control for antigen-specificity, mBECs were loaded with the non-CNS antigen ovalbumin (OVA). Loading of mBECs with OVA did not significantly induce the migration of any of the MOG-specific T-cell subsets, similar to medium-control mBECs (*Figure 3B,C*). However, when mBECs were loaded with myelin, a significant increase in migrated Th1 and Th17 cells was observed, demonstrating that processing and presentation of myelin-derived peptides by mBECs specifically leads to migration of antigen-specific T-cells. Addition of an MHC-II-blocking antibody during the migration period significantly reduced the trans-migration of both Th1 and Th17 cells (*Figure 3D,E*), further providing evidence that presentation of myelin-derived antigens in MHC-II by mBECs facilitates T-cell migration. Using the nominal epitope for 2D2 T-cells (*i.e.* MOG$_{35-55}$) to load mBECs with, similar results were obtained as with myelin-loaded mBECs (*Figure 3F*). Moreover, our observation that OVA-specific Th1 and Th17 only trans-migrated when encountering OVA-loaded BECs and not when co-cultured with MOG$_{35-55}$-loaded or medium control BECs substantiates the finding that T-cell migration over the BEC monolayer occurs in an antigen-specific manner (*Figure 3G,H*). These data demonstrate that brain endothelial cells can internalize antigen and promote antigen-specific T cell transmigration in vitro.

Mice lacking a functional class-II restricted antigen processing machinery are resistant to both active and adoptive transfer EAE (*Tompkins et al., 2002*), suggesting that proper processing of antigens is essential for disease initiation. Although these results could be due to the lack of activation of auto-reactive T-cells by peripheral APCs, the failure to induce disease by adoptive transfer of activated T-cells in this study could also be explained by the lack of a functional antigen processing machinery in BECs since trafficking of injected, ex-vivo activated, T-cells into the brain is impaired.

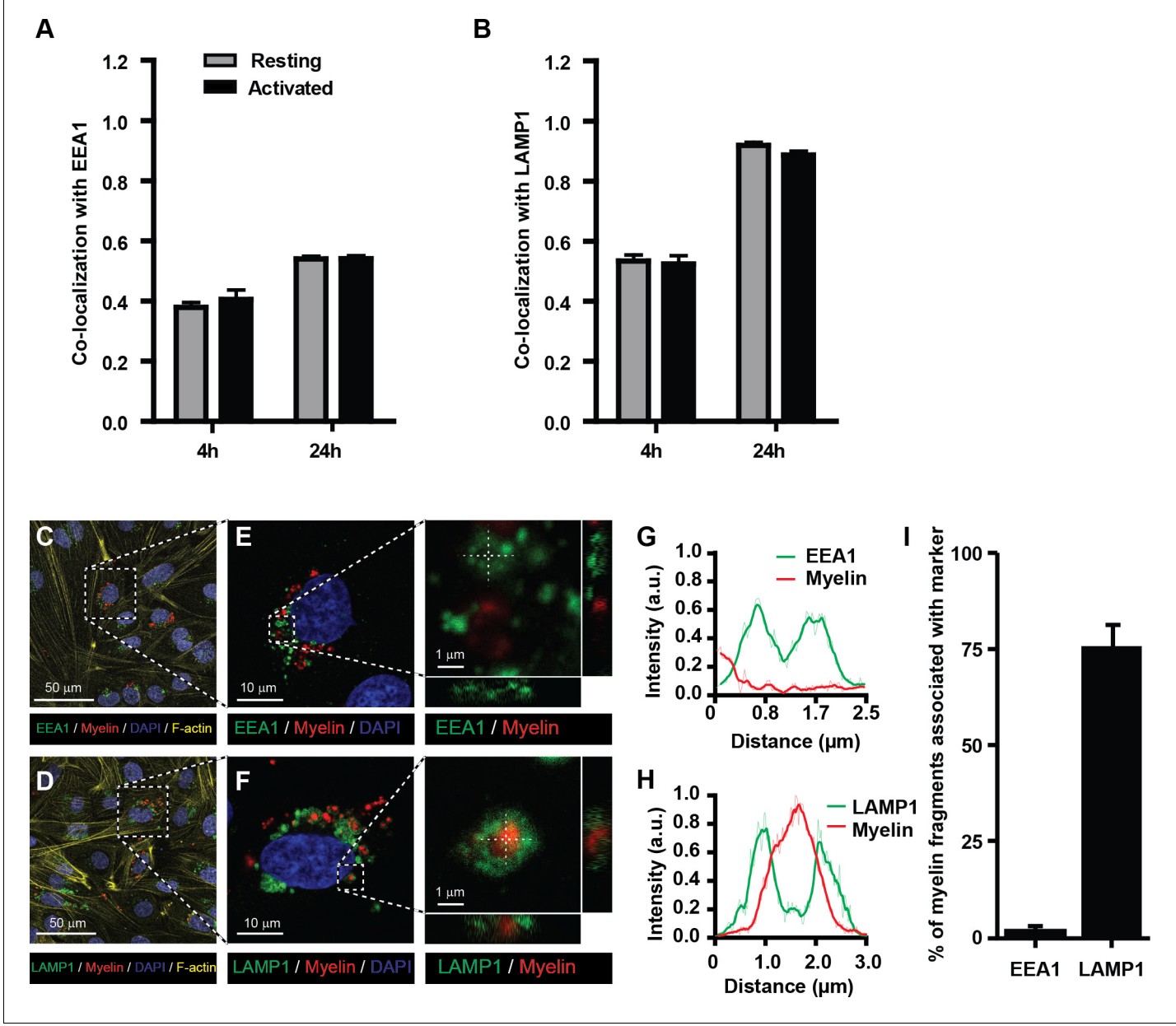

**Figure 2.** Myelin particles are preferably routed to the endo-lysosomes. Resting or activated BECs were loaded with Atto-633 labeled myelin for 4 hr or 24 hr. Uptake of myelin particles and their co-localization with early endosomal (EEA1) or endosomal/lysosomal (LAMP1) compartments was analyzed by imaging flow cytometry and quantified using the brightfield similarity R3 feature (see methods for details). Myelin particles co-localized with (A) EEA1 and (B) LAMP1 in both resting (grey bars) and activated (black bars) BECs. Graphs represent the mean of triplicate values ± SEM of n = 3 independent experiments. (C–G) Adherent BECs were loaded with Atto−633-labeled myelin and 24 hr later, co-localization of myelin (in red) with EEA1 (in green, upper panels) or LAMP1 (in green, lower panels) was analyzed using CSLM. Nuclei were visualized with Hoechst (in blue) and the cytoskeletal F-actin bundles are shown in yellow. Representative images of adherent brain endothelial cells with subcellular localization of myelin with EEA1 (C,E) or LAMP1 (D,F). A magnification of indicated areas is shown in E–F. A cross-sectional study focusing in an myelin-rich area demonstrates the presence of the antigen surrounded by LAMP1 staining, indicating its presence within lysosomes. (G–H) Histograms were created for a selected area (indicated by a line) using ImageJ software (NIH, USA). Histograms were created from each fluorochrome and overlays were made by the program. (I) Quantification of myelin positive early-endosomal and lysosomal compartments. Percentage of myelin fragments associated with each marker was determined using ImageJ software (N = 6).

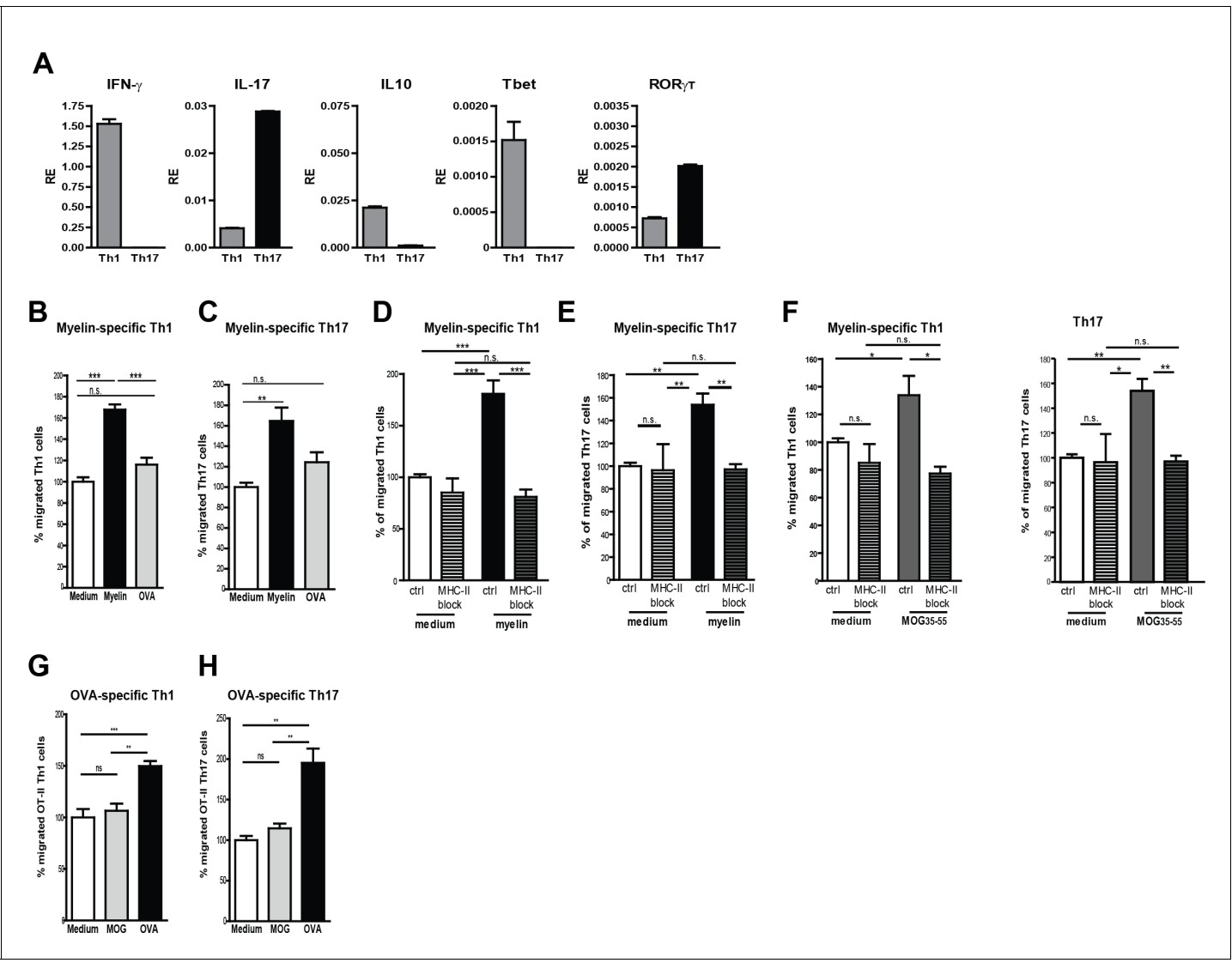

**Figure 3.** Migration of myelin-specific T-cells depends on presentation of myelin-antigens in MHC-II by BECs. (A) Th1 and Th17 subsets were generated in-vitro from naive CD4+CD62L$^{high}$ 2D2 T cells. Expression of IFN-γ, Il−17, IL−10, T-bet and RORγT was determined using qRT-PCR. Data are the means of triplicate values ± SEM of three independent experiments. (B) mBECs were seeded onto trans-wells, activated with TNFα and loaded with myelin for 24 hr. As a control, BECs loaded with the CNS-unrelated antigen OVA or unloaded BECs were used. (B) Th1 or (C) Th17 2D2 T-cells were added to the upper compartment and T-cell migration was quantified by flow cytometry 3 hr later using fluorescent labelled beads as reference. To block antigen recognition by T-cells, an MHC-II blocking antibody was added to mBECs one hour prior addition of the (D) Th1 or (E) Th17 cells. The MHC-II blocking antibody was present during the 3 hr incubation with the T-cells. (F) Transmigration of Th1 and Th17 cells over a monolayer of MOG$_{35-55}$ pulse-loaded activated mBECs was analyzed using Transwells. Migration was assessed in the presence of an MHC-II blocking antibody or control antibody. *p<0.05, **p<0.01, ***p<0.001 (ANOVA with Bonferroni correction). The average frequency of T cells that transmigrated in the control setting are 10.8% ± 1.2 for 2D2 Th1 and 11.6% ± 0.4 for 2D2 Th17. (G,H) Th1 and Th17 subsets were generated in-vitro from naive CD4+CD62L$^{high}$ OT-II T cells. mBECs were seeded onto trans-wells, activated with TNF and loaded with OVA for 24 hr. As a control, BECs loaded with MOG$_{35-55}$ or unloaded BECs were used. Th1 (G) or Th17 (H) OT-II T-cells were added to the upper compartment and T-cell migration was quantified by flow cytometry 3 hr later using fluorescent labelled beads as reference. Average frequency of OT-II Th1 and OT-II Th17 that transmigrated in the control settings are 7.9% ± 1.9 and 12.5% ± 1.4, respectively.**p<0.01, ***p<0.001 (ANOVA with Bonferroni correction).

Thus, together with our novel data, it seems likely that antigen-presentation by the brain endothelium facilitates the entry of antigen-specific CD4+ effector T cells into the brain. This phenomenon has also been proposed, but was never demonstrated, in other diseases suffering from the infiltration and destruction of tissue by auto-reactive T-cells. In Type 1 Diabetes, pancreatic islet antigen

expression was shown to be a key factor in governing the ability of the autoantigen-specific T-cells to accumulate in the pancreatic islets (*Hamilton-Williams et al., 2003*; *Van Halteren et al., 2005*). Importantly, using human T-cell clones and humanized mice, it was demonstrated that only beta cell-specific T-cells reached the pancreatic islets where they destroyed the insulin-producing beta cells. By contrast, diabetes unrelated T-cells retained at the peri-vascular sites (*Unger et al., 2012*), demonstrating that beta-cell-specific T-cells, present in the circulation, need to cross the endothelium to access the pancreatic islets. Whether antigen-specific T-cell entry favors the entry of non-tissue specific T-cells is still a matter of debate. The entry of encephalitogenic T-cells into the brain has been shown to pave the way for non-CNS-specific T-cells (*Lees et al., 2010*; *Ludowyk et al., 1992*) yet the latter subset remained in an inactive state.

Together, the data presented in this study demonstrate for the first time that myelin enters the endosomal/lysosomal pathway when internalized by BECs, irrespective of their activation status. This observation is also different to findings on professional APCs such as dendritic cells, which mainly internalize antigens being in an immature state. The fact that BECs maintain to internalize and process exogenous antigens in an activated state is advantageous during infection-induced inflammation in the brain (*e.g.* meningitis) as it will facilitate the presence of antigen-specific effector T-cells to resolve the unwanted infection. However, this continuous facilitation of immune cell entry into the CNS is destructive in case of MS.

Overall, our results demonstrate that BECs can take up and process myelin particles in a time-dependent manner. Although the focus of the present study was to examine whether antigen-presentation by BECs contributes to transmigration of myelin-reactive T cells, it can be speculated that uptake of myelin, consisting of large particles, by brain endothelial cells predominantly occurs via phagocytosis. BECs have been shown to use different endocytosis mechanisms to internalize particles, which is dependent on the size and composition of the particle (*Faille et al., 2012*; *Falcone et al., 2006*; *Georgieva et al., 2011*). Furthermore, the upregulation of MHC-II expression under inflammatory conditions reinforces the idea of a non-professional antigen presenting cell role. Although we do not provide direct evidence for processing and presentation of internalized myelin, our data strongly suggest that myelin-derived antigens can be presented by brain endothelial cells in MHC-II to antigen-specific T cell subsets, aiding in the diapedesis of these cells in an MHC-II dependent fashion. These results demonstrate that the brain endothelium is an active contributor to disease pathogenesis. Furthermore, these findings have major implications in neuro-inflammatory disorders such as MS, since increased immune cell trafficking has a detrimental effect in disease progression. Therapies directed at antigen processing and presentation by BECs could be effective to dampen unwanted immune cell infiltration in MS.

## Materials and methods

### Cell culture

The human brain endothelial cell (BEC) line hCMEC/D3 (*Weksler et al., 2005*) was kindly provided by Dr PO Couraud (Institut Cochin, Universite Paris Descartes, Paris, France). BECs were grown in EBM−2 medium supplemented with hEGF, hydrocortisone, GA-1000, FBS, VEGF, hFGF-B, R3-IGF-1, ascorbic acid and 2.5% fetal calf serum (Lonza, Basel, Switzerland).

### Flow cytometry

For antigen internalization experiments, resting or 24 hr rhTNFα activated (5 ng/ml, Peprotech, UK) BECs were seeded in collagen-coated plates and when confluent, incubated with 10 µg/ml labeled myelin (myelin−555) for 4 hr or 24 hr. Subsequently, cells were extensively washed with PBS to remove external myelin and fluorescence intensity was measured using a FACS Calibur flow cytometer (Becton and Dickinson, San Jose, CA).

The following antibodies were used to detect the presence of MHC and costimulatory molecules on resting or TNFα activated BECs: FITC-conjugated anti-HLA-ABC (clone DX-17) and -VCAM−1 (clone STA); PE-conjugated anti-HLA-DR (clone G46-6); -CD80 (clone L307.4); -CD86 (clone 2331). Binding of unconjugated anti-CD40 (clone TRAP-1) was detected using goat-anti-mouse IgG1-A488 (Life Technologies). All antibodies were obtained from BD Pharmingen, except anti-VCAM which was obtained from eBiosciences.

## Imaging flow cytometry

Confluent BECs were seeded in 6-well plates (Corning, Amsterdam, The Netherlands) and stimulated with 5 ng/ml rhTNFα for 24 hr. 10 µg/ml of fluorescent-labeled human myelin was added to BECs for 4 hr or 24 hr. Cells were then extensively washed with ice-cold PBS, detached with trypsin and fixated with 4% formaldehyde. Cells were then permeabilized with 0.05% saponin for 30 min at RT and subsequently blocked with 10% goat serum in PBS/BSA. Cells were labeled with EEA1-FITC (BD Bioscience), LAMP1 (BD Pharmingen) and goat anti-mouse Alexa 488 (Molecular Probes, Eugene, OR). Cells were analyzed on the ImageStream X100 (Amnis-Merck Millipore) imaging flow cytometer as previously described (*García-Vallejo et al., 2015*). A minimum of 15,000 cells were acquired per sample. Internalization and co-localization scores were calculated as previously described (*García-Vallejo et al., 2015*). Briefly, cells were acquired on the basis of their area. Analysis was performed with single cells after compensation (with a minimum of 5000 cells). For standard acquisition, the 488 nm laser line (for EEA-1 and LAMP-1) was set at 10 mW and the 642 nm laser line (for myelin) was set at 5 mW.

Firstly, a mask was designed based on the surface of BECs in the brightfield image. This mask was then eroded to exclude the cell membrane. Finally, the resulting mask was applied to the fluorescence channel. The internalization score was then calculated on this mask using the Internalization feature provided in the Ideas v6.0 software (Amnis-Merck Millipore). Internalization can be interpreted as a log-scaled ratio of the intensity of the intracellular space versus the intensity of the entire cell. Cells that have internalized antigen typically have positive scores, while cells that show the antigen still on the membrane have negative scores. Cells with scores around 0 have similar amounts of antigen on the membrane and in intracellular compartments. Co-localization is calculated using the bright detail similarity R3 feature in the Ideas software. This feature corresponds to the logarithmic transformation of Pearson's correlation coefficient of the localized bright spots with a radius of 3 pixels or less within the whole cell area in the two input images. Myelin particle counts were calculated using the peak mask in combination with the spot count feature as previously described (*García-Vallejo et al., 2014*).

## Confocal microscopy

Confluent BECs were seeded in 8-well Ibidi slides (Ibidi, GmbH, Munchen, Germany) and incubated with 10 µg/ml Atto 633 labeled myelin for 24 hr. Subsequently, cells were extensively washed with PBS and fixated with 4% formaldehyde. Non-specific binding was blocked with 5% goat serum in PBS/BSA containing 0.3% Triton-X100. Cells were labeled with rabbit anti-EEA1 (Cell Signaling) or rabbit anti-LAMP1 (Cell Signaling). Antibodies were visualized after 1 hr incubation with goat anti-rabbit Alexa488 (Molecular Probes). Finally, sections were stained with Hoechst (molecular Probes, Invitrogen) to visualize cellular nuclei and with phalloidin rhodamine to visualize F-actin (Molecular Probes, Invitrogen). Sections were mounted with mounting medium. Co-localization was analyzed using a Confocal Laser Scanning Microscope (Leica DMI 6000, SP8, Leica, Mannheim, Germany); images were acquired using LCS software (version 2.61, Leica).

## Isolation and culture of primary murine BECs (mBECs)

Primary mBECs were isolated from brains of C57BL/6 mice as described previously (*Coisne et al., 2005*). Brains were harvested and superficial blood vessel, meninges and cerebellum were removed. Brains were homogenized in isolation medium (HBSS supplemented with 10 mM HEPES and 0.1% BSA) in a potter and centrifuged. The pellet was resuspended in 15% dextran (70 kDa) and centrifuged at 3000 g for 25 min. Subsequently, the pellet was resuspended in 0,2% collagenase/dispase with 10 µg DNase in culture medium (DMEM supplemented with 20% FCS, 1% amino acids, 2% sodium pyruvate and 50 µg/ml gentamycin) and incubated for 30 min in a 37°C waterbath. After washing, the obtained fragments of blood vessels were seeded in collagen-coated dishes in culture medium containing puromycin to avoid contamination with pericytes. After 24 hr of culture, medium was supplemented with 1 ng/ml FGF. At the end of culture, endothelial purity was checked by qPCR for CD31 (endothelial), GFAP (astrocytes), and PDGF-receptor beta (pericytes) as described before (*Reijerkerk et al., 2013*) and cultures were found to be consisting of 95% endothelial cells.

## Generation of MOG-specific T cell subsets

Single cells suspensions of spleens and lymph nodes from 2D2 Tg mice (generous gift from L. Berod, TWINCORE Institute, Hannover, Germany) were depleted of erythrocytes using ACK lysis buffer. Subsequently, CD4$^+$ T-cells were enriched using the mouse CD4$^+$ T-cell enrichment kit (eBiosciences) according to manufacturer's instructions; stained with anti-CD4-PE and CD62L-APC antibodies and naive CD4$^+$CD62L$^{high}$ T cells were sorted using a MoFlow (DakoCytomation, Glostrup, Denmark). Naive T-cells ($5 \times 10^4$) were incubated with MOG$_{35-55}$/LPS loaded BMDCs ($1 \times 10^4$) to promote Th1 differentiation. Incubation of naive CD4$^+$ T-cells with MOG-loaded BMDCs in the presence of PGN (10 µg/ml) promoted Th17 differentiation. Two days later, 10 U/ml rmIL−2 (Invitrogen, Bleijswijk, The Netherlands) was added to the Th1 promoting cultures and another three days later T-cells were harvested and used in functional assays.

## Quantitative PCR

Messenger RNA was isolated from mBECs using the TRIzol method (Life Technologies, Bleiswijk, the Netherlands) and cDNA was synthesized with the Reverse Transcription System kit (Promega, Lei-den, the Netherlands). The following primer sequences were used: IFN-γ FWD: TACTACCTTC TTCAGCAACAGC, IFN-γ REV: AATCAGCAGCGACTCCTTTTC, IL−10-FWD: GGCGCTGTCATCGA TTTCTC; IL−10 REV: ATGGCCTTGTAGACACCTTGG, T-bet FWD: CAGGGAACCGCTTATATG, T-bet REV: CTGGCTCTCCATCATTCA, RORγT FWD: GGAGCAGAGCTTAAACCCCC; RORγT REV: TCCCAGATGACTTGTCCCCA, GAPDH FWD: GACAACTCATCAAGATTGTCAGCA; GAPDH REV: TTCATGAGCCCTTCCACAATG. Oligonucleotides were synthesized by Invitrogen (Bleiswijk, the Netherlands). Quantitative PCR (qPCR) reactions were performed in an ABI7900HT sequence detection system using the SYBR Green method (Applied Biosystems, New York, USA). Expression levels were normalized to GAPDH expression levels.

## Transwell migration

Ex-vivo isolated mBECs were seeded on collagen-coated 5 µm pore size Costar transwells (Corning, Amsterdam, The Netherlands) for 5–7 days. mBECs were loaded with 72.5 µg/ml myelin, 10 µg/ml MOG$_{35-55}$ or 10 µg/ml OVA in the presence of 25 ng/ml TNFα for 24 hr. Cells were thoroughly washed and $1 \times 10^5$ Th1 or Th17 were added per transwell. Anti-mouse MHC-II blocking antibody (#16-5321-81, eBioscience) was added at 5 µg/ml per transwell, 1 hr prior to addition of T cells. After 3 hr T-cells were recovered from the lower well and 20,000 beads (Beckman Coulter, USA) were added to each sample. Samples were analyzed by flow cytometry on a FACScalibur (BD, San Jose, USA) and by gating and counting 5000 beads, the number of migrated T-cells was determined.

## Statistical analysis

Statistical analysis was performed using GraphPad Prism software (v5.01 GraphPad Software, La Jolla, CA) using either unpaired Student t test or one-way ANOVA followed by posthoc Bonferroni correction.

## Acknowledgements

We thank our biotechnicians for excellent care-taking of the animals. This work was supported by the European Research Council (ERCAdvanced339977; WWJU) and the MS research foundation (MS-09-358d, MLP; MS-14-358e, AK).

## Additional information

### Funding

| Funder | Grant reference number | Author |
| --- | --- | --- |
| European Research Council | ERCAdvanced339977 | Wendy WJ Unger |
| Stichting MS Research | MS-09-358d | Melissa A Lopes Pinheiro |
| Stichting MS Research | MS-14-358e | Alwin Kamermans |

The funders had no role in study design, data collection and interpretation, or the decision to submit the work for publication.

## Author contributions

MALP, Designed and performed experiments, Analyzed and interpreted data, Wrote paper, Conception and design, Acquisition of data, Drafting or revising the article; AK, DB, Performed experiments, Analyzed and interpreted data, Acquisition of data; JJG-V, Designed and performed imaging flow cytometric analysis, Analyzed and interpreted data, Conception and design, Acquisition of data; BvhH, LW, Performed experiments, Analyzed data, Acquisition of data; TO, Performed flow cytometric cell sorting, Analyzed and interpreted data, Acquisition of data; MV, Performed experiments, Analyzed and interpreted data, Acquisition of data, Contributed unpublished essential data or reagents; SMAvdP, Acquisition of data; YvK, Designed experiments, Interpreted data, Analysis and interpretation of data; HEdV, Designed experiments, Interpreted data, Wrote paper, Supervised the study, Conception and design, Drafting or revising the article; WWJU, Designed experiments, Interpreted data, Wrote paper, Supervised the study, Conception and design, Acquisition of data, Analysis and interpretation of data, Drafting or revising the article

## Author ORCIDs

Yvette van Kooyk, http://orcid.org/0000-0001-5997-3665
Wendy WJ Unger, http://orcid.org/0000-0001-9484-261X

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
