## [Decision Letter]

Thank you for submitting your work entitled "Internalization and presentation of myelin antigens by the brain endothelium guides antigen-specific T cell migration" for consideration by *eLife*. Your article has been reviewed by two peer reviewers, and the evaluation has been overseen by Gary Westbrook as the Senior Editor. The reviewers have discussed the reviews with one another and the Reviewing Editor has drafted this decision to help you prepare a revised submission. The following individual involved in review of your submission has agreed to reveal his identity: Christopher Carman (peer reviewer).

Summary of essential revisions:

Both reviewers thought this was an important and clinically-relevant topic that has been discussed for some time, but without convincing direct evidence. Both reviewers thought that additional information is required to make the conclusions convincing, but that we are willing to consider a revised manuscript if the critiques can be addressed. The key point in Reviewer 1's review concerns Figure 4. Please also consider the points raised re Figure 1,Figure 2 and given that the results are in vitro, the conclusions in the Discussion should be tempered. The two points in Review 2 may require additional experiments that seem reasonable in scope and potentially possible a 2-month timeframe. In consultation between the editor and the reviewers, we further discussed some of these points that we urge you to address in your revisions:

1) Point 1 of reviewer 2: The issue that the internalization of myelin observed by fluorescence could be due to internalization of membrane as a consequence of cell rounding when they come off their substratum. This issue should really be ruled out by either rigorous co-localization experiments and/or by a control using a fluorescent molecule that sticks to plasma membrane but is not processed by MHCII.

2) Point 2 of reviewer 2: The reviewers were surprised that MHCII is upregulated in ECs by TNFa. Many studies have shown that TNFa increases expression of ICAM-1 and VCAM-1, but not MHCII. In contrast, IFNγ is universally accepted as the major cytokine elevating MHCII in EC. A side-by-side comparison of the two would be instructive and thus the data may be more robust.

3) There is a need to examine internalization on adherent cells and specifically under confluent settings.

Reviewer #1:

Lopes Pinheiro, et al. postulate that endothelial cells (EC) of the blood brain barrier (BBB) present antigen to T cells, which promotes their migration into the brain. They study myelin antigens and EC derived from the CNS in a series of mostly in vitro studies. The authors present data to show that EC from the CNS pulsed with myelin, but not ovalbumin, stimulate the transmigration of myelin-specific 2D2 T cells in Transwell assays. The stimulated transmigration is blocked by antibodies against MHCII. The interpretation is that EC internalize antigen and present it on MHCII to stimulate the antigen-specific T cells. This is an interesting idea, although the concept of antigen presentation by MHCII-expressing endothelial cells has been studied in the setting of transplantation for decades by Jordan Pober's group, and the idea of antigen presentation by BBB EC has been studied, as well, although most of the authors concluded that confluent BBB EC could not successfully present antigen by themselves. (See below). As the data in this manuscript stand, they are consistent with the premise that BBB EC take up myelin, process it in lysosomes, present it to cognate T cells, and stimulate antigen-specific T cell migration. However, the data fall short of direct proof. I do not ask for more experiments. However, I believe better versions of the experiments they have performed are required to prove their point.

1) Figure 1. Could the authors show that the monolayers are indeed confluent? The BBB endothelium is supposed to show very low levels of endocytic activity. How are the antigens taken up and processed? Indeed, several reports show that stimulation of T cells by MHCII expressing BBB EC is far more efficient when the EC are subconfluent than when they are confluent (e.g. J Neuroimmunol. 1995 Sep;61(2):231-9) and that expression of high levels of MHCII by IFNgamma are required for optimum stimulation (J Neuroimmunol. 1999 Jan 1;93(1-2):81-91 and J Neuropathol Exp Neurol. 2000 Feb;59(2):129-36). In contrast, uptake of myelin (Figure 1) and delivery to lysosomes (Figure 2) is said to be independent of cytokine activation in this study. The authors should discuss this.

2) Figure 2. The studies demonstrating delivery to endosomes and lysosomes are difficult to see. Moreover, either the authors have not explained their FRET assay correctly or are not using FRET correctly. In any case, what would be required to show that the EC are processing antigen correctly would be to show that the proper peptides are expressed on MHCII on the surface of these EC in a functional manner. This would ideally require demonstrating that the peptides are expressed on the MHC. This is difficult. I would settle for showing that they can stimulate the antigen-specific proliferation of antigen-specific T cells, not just stimulate migration in a transwell.

3) Figure 3 shows that mouse brain EC in EAE are Class II positive. This has been shown many times before, as far back as 1984 (J Immunol. 1984 May;132(5):2402-7). This figure is nice, but unnecessary, since all the definitive experiments are in vitro.

4) Figure 4 shows that Th1 and Th17 cells generated from MOG-specific 2D2 TCR transgenic mice migrate better across murine EC isolated from the CNS and incubated with myelin than they do across the same EC incubated with ovalbumin (OVA). This does show some biological relevance and is blocked by anti-MHCII antibody. However, the assumption is that the increased migration is due to presentation of cognate antigen by the EC, and this is never demonstrated (see Figure 2). Moreover, the proper control would be to show that either OVA is processed and presented to the same extent by these EC as is myelin and/or to show that OVA-specific T cells would transmigrate the EC pulsed with OVA, but not those pulsed with myelin. The T cells added to the assay have been activated by dendritic cells and expanded with IL-2. They are highly activated. My understanding is that the antigen-specific T cells that would come into the CNS in response to myelin antigens presented by the BBB EC in multiple sclerosis or EAE would be memory cells but not activated at the time but become activated upon presentation of their cognate antigen by the endothelial cells. What happens if the T cells are allowed to rest before being added to the Transwells?

Reviewer #2:

This study investigates an important problem with significant clinical implications. Specifically, the authors examine the underlying basis for auto-reactive myelin-specific inflammatory T cell migration into the brain, a critical event in multiple sclerosis (MS) pathogenesis. Vascular endothelial cells have long been recognized as a type of non-professional antigen presenting cell that may be able to present peptide antigens via MHCII to influence antigen-specific responses in effector/memory CD4^+^ T cells. Studies in MS, and murine models of MS, have suggested that the brain endothelial cells (BEC), which T cells must migrate across in order to enter the brain, may act to selectively recruit myelin-reactive T cells through their ability to present processed myelin peptide via MHCII. However, existing studies have yet to provide direct evidence for such functions. In the current studies the authors demonstrate in vitro, through flow cytometric methods, that BEC are able to constitutively uptake myelin protein where they are targeted to classical MHCII peptide processing compartments (i.e., endosomes and lysosomes). They further show that BEC upregulated MHCII and the co-stimulatory molecule CD40 in response to the inflammatory cytokine TNF-α. The authors go on to demonstrate for the first time that such intact myelin is processes and functionally presented to Th1 and Th17 cells in order to promote their migration across the endothelium in an MHCII-dependent manner. These studies provide new unambiguous evidence that endothelial cells play active roles as non-professional APC in MS pathogenesis and point toward new therapeutic strategies for this disease. This study is generally well performed and presented. However, a couple of important issues/questions remain to be addressed.

1) While the studies to investigate myelin internalization are reasonably well preformed including use of a FRET-based assay to co-localized myelin with endosomal and lysosomal markers. These all use flow cytometric methods (including imaging-based Amnis imaging cytometry). There is some concern that the transition from highly flattened adherent cells to trypsinized cells in suspension (i.e., in order to perform flow cytometry) will almost certainly drive some degree of surface membrane internalization to accommodate the massive reduction in the cell-surface area to volume ratio, that may influence the degree of observed myelin internalization. Thus, this analysis would be strengthened if it also included complementary imaging studies of adherent endothelial cells (e.g., including myelin, EEA1 and/or LAMP1 with Pearson's co-localization analysis).

2) Additionally, the authors show that TNF-α promotes MHCII upregulation on mouse BEC. However, a multitude of studies have shown in human, murine and other endothelial cell types the interferon-γ is the principle driver of endothelial MHCII expression and that TNF-α generally promotes adhesion molecule expression without induction of MHCII. The authors should comment on this as well, ideally as include analysis using interferon-γ.

[Editors' note: further revisions were requested prior to acceptance, as described below.]

Thank you for resubmitting your work entitled "Internalization and presentation of myelin antigens by the brain endothelium guides antigen-specific T cell migration" for further consideration at *eLife*. Your revised article has been favorably evaluated by Gary Westbrook (Senior editor) and two reviewers. The manuscript has been improved but there are some remaining issues that we would like you to address as outlined below before we can make a final determination of acceptance.

Summary:

There were still some concerns about possible limitations of the conclusions. The most important experiment to justify the conclusions has now been included, i.e. the specificity of their transwell assay for the antigen (Figure 3). The authors have shown antigen-specific stimulation of T cell transmigration in vitro across endothelial cells derived from brain in a manner that can be blocked by anti-MHCII antibody. However, the evidence is less convincing that the antigen has been internalized and presented by the endothelial cells, although some of the data are consistent with that hypothesis. A balanced discussion of this issue is important. Please also address these issues:

1) The authors seem to have misunderstood the point of reviewer #2 about their internalization assay. Masking out the surface fluorescence has nothing to do with it. The concern is that when cells go from being flat to round (in this case upon trypsinization) there could be a decrease in cell surface area if membrane is rapidly internalized as vesicles. If these portions of plasma membrane had myelin fragments attached, they would appear as "internalized" even though they were only internalized as part of the procedure. The experiment with flat cells is more convincing. But then, these are not confluent endothelial cells. The was the point 1 of reviewer 2 in the original submission. This concern should be addressed with discussion or with additional data if necessary.

2) One of the reviewers regarded the description of FRET throughout the manuscript and figure as incorrect. The issue was what is described as "FRET" is really illumination of a red fluor by 488nm light, probably as a result of band pass filters. True FRET would be a decrease in green signal due to energy transfer to the red fluor. The authors should either perform this technique correctly or omit this part.

3) The images of "confluent" endothelial cells do show some areas of true confluence with cell borders marked by claudin-5 or VE-cadherin. However, at least half of the monolayers do not show these markers, so they are not truly confluent. Please discuss this issue.

[Editors' note: further revisions were requested prior to acceptance, as described below.]

Thank you for resubmitting your work entitled "Internalization and presentation of myelin antigens by the brain endothelium guides antigen-specific T cell migration" for further consideration at *eLife*. Your revised article has been favorably evaluated by Gary Westbrook (Senior editor) and one of the original reviewers. The manuscript has been improved but there are some remaining issues that need to be addressed as outlined below:

1) Figure 2 now shows internalization and co-localization of myelin with endosomal and lysosomal markers in adherent cells. However, the co-localization shown with LAMP-1 could also be interpreted as a cluster of LAMP-1 positive lysosomes surrounding a myelin-containing endosome. Overlap of the fluorescence should appear yellow. Their images seem to be green blobs surrounding a red one. Because lysosomal diameters are much closer to 1 μm than the 3 μm shown in the scale, the authors should really show an image that shows true overlap of signals rather than what appear in Figure 2 as three separate peaks. Moreover, no quantification of these data (2C-H) are presented, just one set of micrographs for each. Readers need to know what percentage of total myelin fragments were internalized and associated with each marker. The authors should be able to obtain these numbers from their existing data.

2) In Figure 3 the results for transmigration are presented as% of control. The authors should state at least in the figure legend what% of added T cells transmigrated in the control. That will show how robust the assay is.

3) In the subsection “Migration of myelin-specific T-cells depends on presentation of myelin-antigens in MHC-II by BECs”, end of first paragraph: The authors really don't demonstrate that the phenomena of antigen presentation and transmigration are linked, as they admit later in the Discussion. Please reword this sentence to say that the data e.g. "demonstrate that brain endothelial cells can internalize antigen and promote antigen-specific T cell transmigration in vitro".

---

## [Author Response]

*Reviewer #1:*

1) Figure 1. Could the authors show that the monolayers are indeed confluent? The BBB endothelium is supposed to show very low levels of endocytic activity. How are the antigens taken up and processed? Indeed, several reports show that stimulation of T cells by MHCII expressing BBB EC is far more efficient when the EC are subconfluent than when they are confluent (e.g. J Neuroimmunol. 1995 Sep;61(2):231-9) and that expression of high levels of MHCII by IFNgamma are required for optimum stimulation (J Neuroimmunol. 1999 Jan 1;93(1-2):81-91 and J Neuropathol Exp Neurol. 2000 Feb;59(2):129-36). In contrast, uptake of myelin (Figure 1) and delivery to lysosomes (Figure 2) is said to be independent of cytokine activation in this study. The authors should discuss this.

A) The reviewer asks whether the monolayers are indeed confluent (concerning data presented in Figure 1 and Figure 2).

All experiments in this study were performed upon confluency of the brain endothelial monolayers. Within the research team, we have ample knowledge on blood-brain-barrier function studied in confluent monolayers, which mimic blood vessels’ physiological characteristics in vitro. We have extensively tested their characteristics in functional assays as well as performed confocal microscopy analysis on confluent monolayers, as published in van Doorn et al., J Neuroinflam 2012 (doi:10.1186/1742-2094-9-133) and Mizee et al., J Neuroscience 2013 (doi: 10.1523/JNEUROSCI.1338-12.2013). Please see an example of confluent brain endothelial cells as analyzed by confocal microscopy in Figure 4. The brain endothelial cell line hCMEC/D3 we used in our assays is described in Weksler et al., FASEB 2005 (doi:10.1096/fj.04-3458fje).

Author response image 1.Confluent human brain endothelial cells form an impermeable barrier.The human brain endothelial cells (cell line hCMEC/D3) were cultured on Ibidi-slides. After reaching confluence, cells were washed, fixated with 4% formaldehyde and stained with FITC-labeled antibodies specific for A. the tight junction protein Claudin-5 (green). Nuclei were visualized using Hoechst (blue) or B. the adherens junction protein VE-Cadherin (green) and subsequently cells were analyzed by confocal laser scanning microscopy. From the images it is clear that both Claudin-5 and VE-Cadherin are highly present at cell-cell contacts, indicative of tight barrier formation by the brain endothelial cells.**DOI:**
http://dx.doi.org/10.7554/eLife.13149.008

B) How are the antigens taken up and processed?

As shown by many studies, brain endothelial cells use different endocytosis mechanisms to internalize particles: phagocytosis; micropinocytosis and receptor-mediated endocytosis (Faille et al., J Cell Mol Med, 2012; Georgieva et al., Mol therapy 2011; Falcone et al., J. Cell Science 2006). It can be speculated that uptake of myelin, consisting of large particles, by brain endothelial cells predominantly occurs via phagocytosis. However, the aim of our study was to determine whether antigen-presentation by brain endothelial cells contributes to transmigration of myelin-reactive T cells, and not to study the mechanisms used to internalize antigens in detail. We have added this part to the Results & Discussion section (subsection “Migration of myelin-specific T-cells depends on presentation of myelin-antigens in MHC-II by BECs”, last paragraph).

C) The reviewer asks to discuss an apparent discrepancy between the cytokine-dependent increase of MHC-II expression and optimum stimulation of T cells and the cytokine-independent uptake of myelin (Figure 1) and delivery to lysosomes (Figure 2).

Internalization and intracellular routing of antigen are different processes than T cell stimulation and differently influenced by cytokines. Uptake, delivery to lysosomes and subsequent presentation of antigens in MHC molecules also occurs on brain endothelial cells under homeostatic conditions and independent from the presence of inflammatory cytokines (see for example in “Chapter 6: Barriers of the CNS” in Cerebral Circulation, Cipolla MJ., Morgan & Claypool Lifesciences, 2009). However, during inflammation, the processing as well as the presentation of myelin-derived epitopes is increased, as cytokine-mediated activation of the brain endothelial cells results in enhanced expression of MHC-II molecules. Together with enhanced expression of costimulatory molecules (e.g. CD40), this will lead to enhanced T cell activation.

*2) Figure 2. The studies demonstrating delivery to endosomes and lysosomes are difficult to see. Moreover, either the authors have not explained their FRET assay correctly or are not using FRET correctly.*

We apologize to the reviewer for being unclear on how the FRET assay was executed. We have adapted this part of the Materials and methods section with the following:

“Fluorescence resonance energy transfer (FRET) was used as a measure of transfer of energy between the EEA1- or LAMP1-conjugated FITC fluorochromes with the myelin-associated Atto 633 fluorochrome. […] This phenomenon could only be possible if the emitted photons from FITC had excited the atto633-labeled myelin, demonstrating that the proximity of the EEA1/LAMP1 and myelin fluorochromes was close enough to allow FRET.”

Furthermore, the specific images in Figure 2 have been enlarged and the controls are shown as supplementary figure (Figure 2—figure supplement 1) in the manuscript.

In any case, what would be required to show that the EC are processing antigen correctly would be to show that the proper peptides are expressed on MHCII on the surface of these EC in a functional manner. This would ideally require demonstrating that the peptides are expressed on the MHC. This is difficult. I would settle for showing that they can stimulate the antigen-specific proliferation of antigen-specific T cells, not just stimulate migration in a transwell.

We have tested this possibility in co-cultures of in-vitro polarized MOG-specific Th1 or Th17 cells and BECs that were loaded with MOG or myelin in the presence of TNFα. TNFα-activated unloaded BECs were used as controls. When measuring proliferation of the T cells 3 days later, it was clear that the antigen-loaded BECs did not induce proliferation of Th1 nor of Th17 cells. This was found irrespective of using 2D2 or OT-II tg T cells (see Figure 5 showing proliferation of 2D2 Th1 and Th17 cells). However, we hypothesize that this function is likely provided by other cells in the perivascular space, such as the perivascular APCs (Greter et al., Nat Med 2005; Kivisäkk et al., Annals of Neurol 2008; Hickey and Kimura, Science 1988. However, analysis of the activation status of the T cells in these co-cultures revealed that the activation marker CD25 was only expressed when T cells encountered antigen-loaded BECs (see Figure 5). Since induction of CD25 expression is a consequence of TCR signaling (Szamel et al., JI 1998), these data suggest that the epitopes are expressed in MHCII on the surface of BECs in a functional manner.

Author response image 2.Induction of T cell activation but not proliferation by brain endothelial cells.(**A**) Brain endothelial cells were loaded with myelin or MOG35-55 in the presence of TNF and subsequently co-cultured with in vitro generated myelin-specific Th1 or Th17 cells. As a control, none antigen loaded TNF activated brain endothelial cells were used. Proliferation of T cells was determined by incorporation of 3H-Thymidine, which was present during the last 18h of a three day culture period. (**B**) Activation of T cells was determined by analyzing expression of CD25 after 24 hr co-culture with brain endothelial cells that were pulsed with antigen in the presence or absence of TNF prior. Controls included brain endothelial cells that were treated with TNF or medium. Expression of CD25 on T cells was determined using flow cytometry.**DOI:**
http://dx.doi.org/10.7554/eLife.13149.009

3) Figure 3 shows that mouse brain EC in EAE are Class II positive. This has been shown many times before, as far back as 1984 (J Immunol. 1984 May;132(5):2402-7). This figure is nice, but unnecessary, since all the definitive experiments are in vitro.

We agree with the reviewer and have removed the figure from the manuscript.

*4) Figure 4 shows that Th1 and Th17 cells generated from MOG-specific 2D2 TCR transgenic mice migrate better across murine EC isolated from the CNS and incubated with myelin than they do across the same EC incubated with ovalbumin (OVA). This does show some biological relevance and is blocked by anti-MHCII antibody. However, the assumption is that the increased migration is due to presentation of cognate antigen by the EC, and this is never demonstrated (see Figure 2). Moreover, the proper control would be to show that either OVA is processed and presented to the same extent by these EC as is myelin and/or to show that OVA-specific T cells would transmigrate the EC pulsed with OVA, but not those pulsed with myelin.*

We agree with the reviewer and performed these experiments. We observed the same phenomenon when using antigen-loaded BECs and OT-II T cells as for the 2D2 T cells: OVA-specific T cells only transmigrated when encountering OVA-loaded BECs and did not when cultured with MOG-loaded or none antigen-loaded BECs (see Figure 6). These data have been added to the manuscript (Results & Discussion section, subsection “Migration of myelin-specific T-cells depends on presentation of myelin-antigens in MHC-II by BECs”, first paragraph and as Figure 3—figure supplement 1).

Author response image 3.OVA-specific T-cells only transmigrate upon encounter of BECs presenting OVA-antigens in MHC-II.Th1 and Th17 subsets were generated in-vitro from naive CD4^+^CD62L^high^ OT-II T cells. mBECs were seeded onto trans-wells, activated with TNF and loaded with OVA for 24h. As a control, BECs loaded with MOG_35-55_ or unloaded BECs were used. Th1 (left graph) or Th17 (right graph) OT-II T-cells were added to the upper compartment and T cell migration was quantified by flow cytometry 3h later, using fluorescent labelled beads as reference. **p<0.01, ***p<0.001 (ANOVA with Bonferroni correction).**DOI:**
http://dx.doi.org/10.7554/eLife.13149.010

The T cells added to the assay have been activated by dendritic cells and expanded with IL-2. They are highly activated. My understanding is that the antigen-specific T cells that would come into the CNS in response to myelin antigens presented by the BBB EC in multiple sclerosis or EAE would be memory cells but not activated at the time but become activated upon presentation of their cognate antigen by the endothelial cells. What happens if the T cells are allowed to rest before being added to the Transwells?

To differentiate the naive T cells into Th1 or Th17 cells as well as to expand them, the T cells were activated in vitro by antigen-loaded DCs. However, at time of addition to the antigen-loaded BECs, which is 7 days after the initial stimulus, the T cells were rounded up again and receptive for re-stimulation. Moreover, upon activation these cells secreted significant amounts of effector cytokines (as measured by intracellular staining and flow cytometry), which would be hampered if they would still be activated.

*Reviewer #2:*

*This study is generally well performed and presented. However, a couple of important issues/questions remain to be addressed.*

1) While the studies to investigate myelin internalization are reasonably well preformed including use of a FRET-based assay to co-localized myelin with endosomal and lysosomal markers. These all use flow cytometric methods (including imaging-based Amnis imaging cytometry). There is some concern that the transition from highly flattened adherent cells to trypsinized cells in suspension (i.e., in order to perform flow cytometry) will almost certainly drive some degree of surface membrane internalization to accommodate the massive reduction in the cell-surface area to volume ratio, that may influence the degree of observed myelin internalization. Thus, this analysis would be strengthened if it also included complementary imaging studies of adherent endothelial cells (e.g., including myelin, EEA1 and/or LAMP1 with Pearson's co-localization analysis).

In order to measure whether the localization of the myelin signal was intracellular or membrane-bound, we designed a mask that excludes the cell membrane and calculated a ratio of the amount of fluorescence located in the mask vs the total amount of fluorescence, as previously reported (Garcia-Vallejo et al., Front Immunol 2015). The results indicate that the myelin fluorescence signal was intracellular, demonstrating that BECs are able to efficiently internalize myelin (Figure 7). Using Imaging flow cytometry we could clearly see that myelin was internalized and not located on the membrane. However, the reviewer is correct that if there was myelin attached to the membrane but not internalized at the moment of trypsinization, this signal would be gone and would be undetectable as not-internalized material. We therefore performed, as suggested, experiments on adherent BECs. Using confocal laser-scanning microscopy (CSLM) we ascertained that myelin is indeed internalized by the BECs and does not remain on the membrane. Furthermore, it is clear that 24 hours after loading, internalized myelin is contained within LAMP1 positive vesicles and not with EEA1 positive organelles (Author response image B-G), which corroborate our observations made with Imaging flow cytometry.

Author response image 4.Myelin is internalized and routed to lysosomal compartments by brain endothelial cells.(**A**) BECs were loaded with Atto 633-labeled human myelin for 24h and internalization of myelin particles was assessed by imaging flow cytometry. To determine internalization scores, a mask was designed based on the surface of BECs in the brightfield image. This mask was then eroded to exclude the cell membrane. The resulting mask was applied to the fluorescence channel. The internalization score, interpreted as a ratio of the intensity of the intracellular space versus the intensity of the whole cell, was calculated on this mask using the internalization feature of the Ideas v6.0 software (AMNIS Merck Millipore). Cells that have internalized antigens have positive scores, as depicted here for BECs. (**B-G**) Adherent brain endothelial cells were incubated with Atto-633-labeled myelin and 24h later, co-localization of myelin (Red) with early endosomal (EEA1, Green, upper panels) or endosomal/lysosomal (LAMP1, Green, lower panels) compartments was analyzed using CSLM. Nuclei were visualized with Hoechst (blue). Representative images of adherent brain endothelial cells with subcellular localization of Myelin with EEA1 (or LAMP1 (**C**). A magnification of indicated areas is shown in D B)+E. F+G. Histograms were created for a selected area (indicated by a line) using ImageJ software (NIH, USA). Histograms were created from each fluorochrome and overlays were made by the program. Histograms clearly show the lack of association of myelin with early endosomes but the enclosure of myelin within LAMP1 positive vesicles.**DOI:**
http://dx.doi.org/10.7554/eLife.13149.011

The Imaging flow cytometry data showing the internalization score of myelin have been added to the manuscript (Results, subsection “Brain endothelial cells internalize exogenous antigens irrespective of their activation status”, last paragraph and as Figure 1—figure supplement 1).

The new CLSM data obtained on the internalization and intracellular routing of myelin in adherent brain endothelial cells are added to the Results & Discussion section (subsection “Myelin internalized by BECs is directed to the endo-lysosome compartments”, first paragraph) and as Figure 2—figure supplement 2.

*2) Additionally, the authors show that TNF-α promotes MHCII upregulation on mouse BEC. However, a multitude of studies have shown in human, murine and other endothelial cell types the interferon-γ is the principle driver of endothelial MHCII expression and that TNF-α generally promotes adhesion molecule expression without induction of MHCII. The authors should comment on this as well, ideally as include analysis using interferon-γ.*

We have additionally activated brain endothelial cells with IFNγ and analyzed the expression of MHC-II as well as MHC-I, VCAM-1 and CD40 using flow cytometry. As shown in Figure 6, IFN-γ increases, similar to TNF, expression of all markers analyzed. Although the expression of MHC-I, VCAM-1 and CD40 seems to be more enhanced by IFN-γ than by TNFα, in our hands both cytokines increase expression of these costimulatory and MHC molecules.

Author response image 5.Human brain endothelial cells increase MHC and costimulatory molecule expression upon activation by IFN-γ.Confluent monolayers of brain endothelial cells (BECs) were stimulated with 10 ng/ml IFN-γ for 24 hr. Expression of MHC-I, MHC-II, CD40 and VCAM-1 was determined by flow cytometry. Histograms depict expression of indicated markers in resting (orange line) and activated (green line) BECs. Red and blue lines indicate isotype controls.**DOI:**
http://dx.doi.org/10.7554/eLife.13149.012

[Editors' note: further revisions were requested prior to acceptance, as described below.]

Summary:

1) The authors seem to have misunderstood the point of reviewer #2 about their internalization assay. Masking out the surface fluorescence has nothing to do with it. The concern is that when cells go from being flat to round (in this case upon trypsinization) there could be a decrease in cell surface area if membrane is rapidly internalized as vesicles. If these portions of plasma membrane had myelin fragments attached, they would appear as "internalized" even though they were only internalized as part of the procedure. The experiment with flat cells is more convincing. But then, these are not confluent endothelial cells. The was the point 1 of reviewer 2 in the original submission. This concern should be addressed with discussion or with additional data if necessary.

We understand the concerns from the reviewers regarding the confluency of the endothelial cells.

Both claudin-5 and VE-cadherin are highly present at cell-cell contacts in the confluent human brain endothelial cells, indicative of tight barrier formation by the brain endothelial cells. We agree that there appear to be differences in the expression level of claudin-5 and VE-cadherin at the cell-cell border throughout the monolayers, which may be the result of the presence of serum upon the culturing. To demonstrate that the brain endothelial cells form confluent monolayers we performed additional staining for the cytoskeletal F-actin, which reveals its localization of the cell-cell contacts, indicative of confluency (Figure 9).

Author response image 6.Confluency of cultured brain endothelial cells shown using F-actin, Claudin-5 and VE-Cadherin staining on samples shown in Figure 2 of the manuscript.**DOI:**
http://dx.doi.org/10.7554/eLife.13149.013

2) One of the reviewers regarded the description of FRET throughout the manuscript and figure as incorrect. The issue was what is described as "FRET" is really illumination of a red fluor by 488nm light, probably as a result of band pass filters. True FRET would be a decrease in green signal due to energy transfer to the red fluor. The authors should either perform this technique correctly or omit this part.

We performed additional FRET experiments using a more conventional setup to study FRET, i.e. FilterFRET (Figure 2) and acceptor photo-bleaching (Figure 2), by confocal laser scanning microscopy on adhered brain endothelial cells. Using this approach, we could not show the occurrence of FRET in our studies. We therefore agree with and sincerely apologize to the reviewers that the FRET as described previously (measured using imaging flow cytometry), is no “true” FRET, revealing the limitations of this techniques (possibly due to the lower resolution in comparison to the CLSM). However, this technique remains suitable to demonstrate the colocalization between myelin particles and EEA1 or LAMP1. We have adapted Figure 2 and the respective Results section in the manuscript accordingly.

*3) The images of "confluent" endothelial cells do show some areas of true confluence with cell borders marked by claudin-5 or VE-cadherin. However, at least half of the monolayers do not show these markers, so they are not truly confluent. Please discuss this issue.*

See response to point #1.

[Editors' note: further revisions were requested prior to acceptance, as described below.]

*The manuscript has been improved but there are some remaining issues that need to be addressed as outlined below:*

*1) Figure 2 now shows internalization and co-localization of myelin with endosomal and lysosomal markers in adherent cells. However, the co-localization shown with LAMP-1 could also be interpreted as a cluster of LAMP-1 positive lysosomes surrounding a myelin-containing endosome. Overlap of the fluorescence should appear yellow. Their images seem to be green blobs surrounding a red one.*

We understand the concerns about the co-localization between LAMP1 and Myelin. For visualization of the lysosomes, we used LAMP1, which is a transmembrane protein of the lysosome. We thus stain the membrane of lysosomes. The myelin that has been taken up by the lysosomes is inside the lumen of the lysosome. This results in the observed images where we see myelin surrounded by LAMP1 staining. To show that this is also the case in the Z-direction, we included orthogonal views of the lysosomes in Figure 2 of the manuscript.

Due to the optical sectioning of the confocal microscope the LAMP1 positive lysosomes are observed as ring shaped lysosomes, at which the intensity of the Lamp1 staining is very bright at the lysosomal membrane, while in the lumen the intensity drops significantly. The intensity of myelin is very high in the lumen of the lysosome. Due to these difference in intensity, the overlap of the fluorescence does not appear yellow. Yellow overlap only occurs when the fluorescence intensities of both channels is equal. To illustrate this, we generated green and red images of different intensities and their corresponding merge images and histograms (see Figure 10).

Author response image 7.Co-localization of green and red channels.(**A**) co-localization of high intensity green and high intensity red image shows a yellow merge image. Histogram shows overlap of intensities. (**B**) Co-localization of low intensity green dot with bright green ring with a high intensity red dot, Merge image does not show yellow colour. Histogram shows high intensity of green channel at the ring and high intensity of red channel at the inside. (**C**) Green LAMP1 positive lysosome shows high intensity border and lower intensity lumen together with high intensity myelin in red. The merge image does not appear yellow.**DOI:**
http://dx.doi.org/10.7554/eLife.13149.014

*Because lysosomal diameters are much closer to 1 μm than the 3 μm shown in the scale, the authors should really show an image that shows true overlap of signals rather than what appear in Figure 2 as three separate peaks.*

The shown lysosome is indeed relative large. However, it has been reported that the size of a lysosome differs depending on cell type, and in most cells lysosomes can be up to 1.5 μm in diameter (Lüllmann-Rauch R. History and morphology of the lysosome. In: Saftig P., editor.Lysosomes. New York: Springer; 2005). It has also been reported that lysosomes increase in size as a result of accumulation of undigested material (Appelqvist et al., J Mol Cell Biol. 2013). To support this statement, we measured the size of myelin containing (LAMP1-positive) lysosomes as well as none myelin containing lysosomes. The average size of myelin containing lysosomes is 1.4 ± 0.08 µm while the size of none myelin containing lysosomes is around 1.1 ± 0.1 µm (*P*=0.004; N=10; please see Figure 10).

The image in the previous Figure 2 of the manuscript showed a lysosome with a size of 1.8 µm, we therefore exchanged the image for another image that more closely represents the average measured lysosome size.

Author response image 8.Average size of lysosomes in myelin-loaded brain endothelial cells.The diameter of lysosomes was determined using ImageJ software and shown as histograms. The average size of 5 lysosomes per cell of a total of 10 cells that contain myelin is compared with that of none myelin containing lysosomes and represented in the bar graph.**DOI:**
http://dx.doi.org/10.7554/eLife.13149.015

Moreover, no quantification of these data (2C-H) are presented, just one set of micrographs for each. Readers need to know what percentage of total myelin fragments were internalized and associated with each marker. The authors should be able to obtain these numbers from their existing data.

We apologize for not providing the quantification of the CSLM data. Using both CSLM and imaging flow cytometry (ISX) we find that approximately 50% of brain endothelial cells contain myelin. Moreover, the majority of the myelin is present in LAMP1 positive lysosomes.We have added this quantification as panel I to Figure 2 in the manuscript. The data obtained by ISX were already included in Figure 1 of the manuscript.

2) In Figure 3 the results for transmigration are presented as% of control. The authors should state at least in the figure legend what% of added T cells transmigrated in the control. That will show how robust the assay is.

The average frequency of T cells that transmigrated in the control setting are 10.8% ± 1.2 for 2D2 Th1; 11.6% ± 0.4 for 2D2 Th17; 7.9% ± 1.9 for OT-II Th1 and 12.5% ± 1.4 for OT-II Th17. The indicated percentages of migrated control cells have been added to the legend of Figure 3.

Although these percentages may seem low in comparison to other studies on endothelial transmigration by T cells, we want to emphasize that these studies used cell lines (of both brain endothelial cells as well as of T cells), in contrast to our study using primary brain endothelial cells and primary T cells. Of note, in a study by Reiss and Engelhardt (Int Immunol. 1999) similar frequencies of transmigrated T cells in the controls were found when using primary T cells.

*3) In the subsection “Migration of myelin-specific T-cells depends on presentation of myelin-antigens in MHC-II by BECs”, end of first paragraph: The authors really don't demonstrate that the phenomena of antigen presentation and transmigration are linked, as they admit later in the Discussion. Please reword this sentence to say that the data e.g. "demonstrate that brain endothelial cells can internalize antigen and promote antigen-specific T cell transmigration in vitro".*

We have adapted this sentence as requested.